# Splines Parameterization of Planar Domains by Physics-Informed Neural Networks

Antonella Falini [1], Giuseppe Alessio D'Inverno [2], Maria Lucia Sampoli [2] and Francesca Mazzia [1,*]

[1] Department of Computer Science, University of Bari, 70125 Bari, Italy; antonella.falini@uniba.it
[2] Department of Information Engineering and Mathematics, University of Siena, 53100 Siena, Italy; dinverno@diism.unisi.it (G.A.D.); marialucia.sampoli@unisi.it (M.L.S.)
[*] Correspondence: francesca.mazzia@uniba.it

**Abstract:** The generation of structured grids on bounded domains is a crucial issue in the development of numerical models for solving differential problems. In particular, the representation of the given computational domain through a regular parameterization allows us to define a univalent mapping, which can be computed as the solution of an elliptic problem, equipped with suitable Dirichlet boundary conditions. In recent years, Physics-Informed Neural Networks (PINNs) have been proved to be a powerful tool to compute the solution of Partial Differential Equations (PDEs) replacing standard numerical models, based on Finite Element Methods and Finite Differences, with deep neural networks; PINNs can be used for predicting the values on simulation grids of different resolutions without the need to be retrained. In this work, we exploit the PINN model in order to solve the PDE associated to the differential problem of the parameterization on both convex and non-convex planar domains, for which the describing PDE is known. The final continuous model is then provided by applying a Hermite type quasi-interpolation operator, which can guarantee the desired smoothness of the sought parameterization. Finally, some numerical examples are presented, which show that the PINNs-based approach is robust. Indeed, the produced mapping does not exhibit folding or self-intersection at the interior of the domain and, also, for highly non convex shapes, despite few faulty points near the boundaries, has better shape-measures, e.g., lower values of the Winslow functional.

**Keywords:** physics-informed neural networks; planar domains; quasi-interpolation; spline parameterization

**MSC:** 65D07; 65D17; 65N50

## 1. Introduction

Computer-Aided Design (CAD) systems only provide the boundary representation of the given computational domain, but in order to perform numerical simulation, a representation of the interior is often necessary. While many techniques based on the use of triangulations, see, e.g., [1,2], are available for unstructured grids, the generation of structured grids via analysis-suitable parameterizations is still challenging, especially when the considered domains are not convex. The main requirement is to obtain a bijective, and, hence, a folding-free, mapping $\mathcal{M}$ defined on the reference domain $\widehat{\Omega} = [0,1]^2$ which provides a description of the considered computational domain $\Omega$,

$$\mathcal{M} \quad : \quad \widehat{\Omega} \to \Omega$$
$$\forall (t_1, t_2) \in \widehat{\Omega}, \exists (\mathbf{x}, \mathbf{y}) \in \Omega : \mathcal{M}(t_1, t_2) \quad = \quad (\mathbf{x}(t_1, t_2), \mathbf{y}(t_1, t_2)). \tag{1}$$

The image of a uniform, Cartesian grid in $\widehat{\Omega}$ under the mapping $\mathcal{M}$ is a curvilinear, boundary conforming grid in the physical domain $\Omega$, with a uniform topological structure, i.e., same number of vertices, cells, neighboring cells, and so on. When the final goal is to

perform numerical simulation on $\Omega$, by using the pull-back of the mapping $\mathcal{M}$, the considered differential problem can be efficiently carried out on the parametric domain $\widehat{\Omega}$, see, e.g., [3,4]. Hence, the invertibility requirement imposed on $\mathcal{M}$ becomes fundamental.

From the computational point of view, the cheapest techniques to construct $\mathcal{M}$ rely on transfinite interpolation, such as the Coons patches [5] and the spring model [6]. Unfortunately, such techniques fail to produce a folding-free $\mathcal{M}$ when the assayed domain $\Omega$ has a complex shape or is not convex. Therefore, other more advanced methods, based on the use of specific functionals that control the quality of the obtained parameterization [7–9], are adopted, together with the use of conformal and harmonic mappings which intrinsically ensure a high-quality result, see, e.g., [10–12]. In the present paper, an approach based on the so-called Elliptic Grid Generation (EGG) methods is proposed. Indeed, EGG methods are particularly suitable for the current setting as only a description of the boundary of $\Omega$ is required. The mapping $\mathcal{M}$ is constructed as the solution of a system of elliptic PDEs defined on $\widehat{\Omega}$ subject to Dirichlet boundary conditions, i.e., $\mathcal{M}(\partial\widehat{\Omega}) = \partial\Omega$. Moreover, the bijectivity of $\mathcal{M}$ is guaranteed as long as the numerical accuracy is good enough and the produced curvilinear grid in $\Omega$ is smooth, which results in small truncation errors when domain methods are employed to perform numerical simulation on the considered $\Omega$. We refer to [13–15] and references therein for a comprehensive review of EGG techniques and their recent applications. When the shape of $\Omega$ is rather challenging or not convex, a segmentation in subdomains, called patches, with easier shape is usually advised and the description of $\Omega$ is thus obtained as an atlas whose charts are the individual bijective parameterizations for each patch. The main challenges of such approach consist in the identification of a suitable segmentation technique [16–19] and in the smooth transition between the patches [20,21].

In the last two decades, machine learning and deep learning techniques have started to play an active role in the setting up of new methods for the numerical solution of PDEs [22–24]. In particular, Physics-Informed Neural Networks (PINNs) [25–27] have emerged as an intuitive and efficient deep learning framework to solve PDEs, carrying on the training of a neural network by minimizing the loss functional which incorporates the PDE itself, *informing* the neural network about the physical problem to be solved.

In the proposed method, the final parameterization is obtained by means of quasi-interpolation; a local approach to construct approximants to given functions or data with full approximation order. Spline quasi-interpolants (QI) are usually defined as linear combination of locally supported basis functions forming a convex partition of unity. The coefficients of such a linear combination, called functionals, can be defined in different ways by taking into account the function evaluations, its derivative information or its integral values, see for example [28–31]. In the present paper we rely on the QI-Hermite technique introduced in [32], which is a differential type QI used together with PINNs to produce a single-patch parameterization methodology. In particular, a suitable PINNs architecture and loss function will be presented to fit within the EGG methods framework. Finally, the use of the QI operator will allow us to produce a robust output regardless from the input shape of $\Omega$ without requiring a preliminary segmentation of the computational domain. The main contribution of this work consists of introducing a novel algorithm to compute a single patch planar domain parameterizations. In particular:

- The discrete description of the computational domain is achieved by using PINNs.
- The continuous representation of the computational domain is then obtained by using a suitable QI operator which provides a spline parameterization, i.e., a continuous description, of the desired smoothness.

The paper is organized as follows. Section 2 summarizes the main concepts about quasi-interpolation and PINNs; Section 3 is devoted to the pipeline description of our method; then numerical examples are presented in Section 4 in order to analyze the quality of the proposed method and in Section 5 a suitable post-processing step is described in order to handle more challenging benchmarks. Finally, Section 6 draws some conclusive remarks.

## 2. Preliminaries

Let us summarize the main concepts of the adopted QI scheme together with the basics ideas of the PINNs model.

In the following, the spline Hermite QI introduced in [32] is considered. In particular, its version based on the derivative approximation presented in [33] will be employed. For the univariate case, let $S_{p,T}$ be the space of splines with degree $p$ and associated extended knot vector $T$ defined in the reference domain $[0,1]$; $T = \{t_0 \leq \cdots \leq t_{p-1} \leq t_p < \cdots < t_{m+1} \leq \cdots \leq t_{m+p+1}\}$, with $t_p = 0$ and $t_{m+1} = 1$. A spline $s \in S_{p,T}$ can be represented by using the standard B-spline basis $B_{j,p}, j = 0, \ldots, m$, defined on $T$:

$$s(\cdot) = \sum_{j=0}^{m} \mu_j B_{j,p}(\cdot).$$

The unknown coefficient vector $\boldsymbol{\mu} := (\mu_0, \ldots, \mu_m)^T$ can be computed by solving local linear systems of dimension $2p \times 2p$. The derivatives are approximated by using a symmetric finite difference scheme and, hence, the entries of the vector $\boldsymbol{\mu}$ are a linear combination of the function to be approximated.

The tensor product formulation of the scheme can be easily derived; a spline $s$ in the space $S_{p_1,T_1} \times S_{p_2,T_2}$, can be written as,

$$s(t_1, t_2) = \sum_{i=0}^{m_1} \sum_{j=0}^{m_2} \mu_{i,j} B_{i,p_1}(t_1) B_{j,p_2}(t_2).$$

Setting $\mathbf{t} := (t_1, t_2)$, $\mathbf{p} := (p_1, p_2)$ and $\mathbf{I} := \{(i,j), i = 0, \ldots, m_1, j = 0, \ldots, m_2\}$, $s$ can be expressed compactly as,

$$s(\mathbf{t}) = \sum_{\mathbf{i} \in \mathbf{I}} \mu_{\mathbf{i}} \mathcal{B}_{\mathbf{I},\mathbf{p}}(\mathbf{t}), \tag{2}$$

with $\mathcal{B}_{\mathbf{I},\mathbf{p}}(\mathbf{t}) := B_{i,p_1}(t_1) B_{i,p_1}(t_2)$ tensor product B-spline basis. The approximation order of the scheme is maximal if the approximation order of the derivatives is bigger than $\mathbf{p} - 1$. For more technical details we refer to [34].

PINNs are a class of learning algorithms used to solve problems involving PDEs [25,27]. An $L$-layers neural network (typically, a feedforward neural network) $\mathcal{N}^L(\mathbf{x}) : \mathbb{R}^{\mathbf{d_{in}}} \to \mathbb{R}^{\mathbf{d_{out}}}$ is an architecture consisting of $N_\ell$ neurons at the $\ell$-th layer with $N_0 = \mathbf{d_{in}}$ and $N_L = \mathbf{d_{out}}$. Let $\mathbf{W}^\ell \in \mathbb{R}^{\mathbf{N}_\ell \times \mathbf{N}_{\ell-1}}$ and $\mathbf{b}^\ell \in \mathbb{R}^{\mathbf{N}_\ell}$ be, respectively, the matrix of weights and the vector of bias at the layer $\ell$. The net $\mathcal{N}^L$ then can be generally described with the following scheme:

$$
\begin{aligned}
\mathcal{N}^0(\mathbf{x}) &= \mathbf{x} \in \mathbb{R}^{\mathbf{d_{in}}} \text{ input layer} \\
\mathcal{N}^\ell(\mathbf{x}) &= \sigma\left(\mathbf{W}^\ell \mathcal{N}^{\ell-1}(\mathbf{x}) + \mathbf{b}^\ell\right) \in \mathbb{R}^{N_\ell} \ \ell = 1, \ldots, L-1 \text{ hidden layers} \\
\mathcal{N}^L(\mathbf{x}) &= \mathbf{W}^L \mathcal{N}^{L-1}(\mathbf{x}) + \mathbf{b}^L \in \mathbb{R}^{\mathbf{d_{out}}} \text{ output layer,}
\end{aligned}
$$

with $\sigma$ a non linear activation function. The constructed net is trained to compute the (approximate) solution of the involved PDE, and the training phase is carried out by minimizing a suitable loss functional which takes into account the given boundary conditions along with the so-called *residual term*. For a theoretical investigation of the convergence properties and stability analysis related to PINNs, see, e.g., [35,36].

## 3. The Method

The aim of the proposed EGG-based method consists in computing a parameterization $\mathcal{M}$ of a given planar domain $\Omega$ by solving the following system,

$$
\begin{cases}
-\Delta \mathcal{M} = 0 & \text{in } \Omega \\
\mathcal{M} = g_i & \text{on } \Gamma_i, \ i = 1, \ldots, 4,
\end{cases} \tag{3}
$$

where the boundary of the domain $\partial\Omega = \bigcup_{i=1}^{4} \Gamma_i$ can be described as the non-overlapping union of four boundary curves and the Dirichlet boundary conditions are given by the functions $g_i : [0,1] \to \Gamma_i\ i = 1, \dots, 4$ invertible parameterizations of the boundary curves, according to the scheme exemplified in Figure 1.

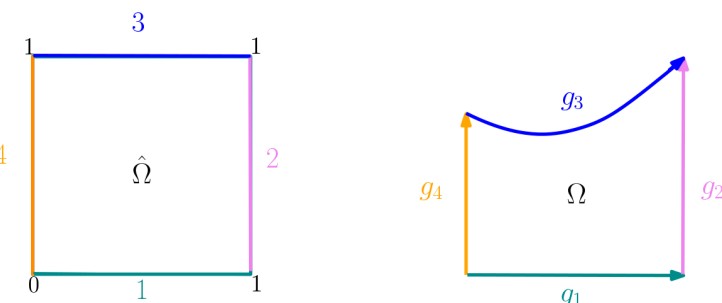

**Figure 1.** Scheme of the applied Dirichlet boundary conditions.

A net $\mathcal{N}^L$ consisting of $L = 6$ layers with $\mathbf{d_{in}} = \mathbf{d_{out}} = 2$ and for $\ell = 1, \dots, L-2$, $N_\ell = 100$ neurons is constructed. Then, the provided output is an approximate solution $\widehat{\mathcal{M}}_{\boldsymbol{\theta}}(\mathbf{t})$ of problem (3) which depends on a set of parameters $\boldsymbol{\theta} = \{\mathbf{W}^\ell, \mathbf{b}^\ell\}_{0 \le \ell \le L}$. The following functional is minimized,

$$\mathcal{L}(\mathcal{T}, \boldsymbol{\theta}) = \mathcal{L}_\Gamma(\mathcal{T}_\Gamma, \boldsymbol{\theta}) + w\mathring{\mathcal{L}}(\mathring{\mathcal{T}}, \boldsymbol{\theta}), \tag{4}$$

where $\mathcal{T}_\Gamma$ collects uniformly sampled parameters $t_i$ corresponding to points located on the four boundary curves, while $\mathring{\mathcal{T}}$ is a set of uniformly sampled points inside $\widehat{\Omega}$. More precisely, the loss functional in (4) consists of the terms,

$$\begin{aligned}
\mathcal{L}_\Gamma(\mathcal{T}_\Gamma, \boldsymbol{\theta}) \quad &:= \quad \frac{1}{|\mathcal{T}_\Gamma|} \sum_{t_1, t_2 \in \mathcal{T}_\Gamma} \Big( (\widehat{\mathcal{M}}_{\boldsymbol{\theta}}(t_1, 0) - g_1(t_1))^2 + (\widehat{\mathcal{M}}_{\boldsymbol{\theta}}(1, t_2) - g_2(t_2))^2 + \\
&\qquad (\widehat{\mathcal{M}}_{\boldsymbol{\theta}}(t_1, 1) - g_3(t_1))^2 + (\widehat{\mathcal{M}}_{\boldsymbol{\theta}}(0, t_2) - g_4(t_2))^2 \Big), \\
\mathring{\mathcal{L}}(\mathring{\mathcal{T}}, \boldsymbol{\theta}) \quad &:= \quad \frac{1}{|\mathring{\mathcal{T}}|} \sum_{\mathbf{t} \in \mathring{\mathcal{T}}} (-\Delta\widehat{\mathcal{M}}_{\boldsymbol{\theta}}(\mathbf{t}))^2,
\end{aligned}$$

where the first term $\mathcal{L}_\Gamma(\mathcal{T}_\Gamma, \boldsymbol{\theta})$ corresponds to the mean squared error between the predicted location of the boundary points and the assigned points given by the boundary conditions, while the second term $\mathring{\mathcal{L}}(\mathring{\mathcal{T}}, \boldsymbol{\theta})$ is called *residual term* and relates directly to the differential problem (this explains why the neural network is called *physics informed*). Its computation is carried on by means of *automatic differentiation*, a procedure to compute derivatives with respect to sample batches based on backpropagation technique [37], nowadays implemented in most of Deep Learning packages, such as `Tensorflow` [38], `PyTorch` [39], and `Jax` [40]. The parameter $w$ in (4) is determined via thresholding on the proximity of the interior points to the boundaries; $w = 0$ if $\text{dist}(\widehat{\mathcal{M}}_{\boldsymbol{\theta}}(\mathbf{t}), \partial\Omega) < 10^{-4}$, otherwise $w = 1$. Training is performed on an Intel(R) Core(TM) i7-9800X processor running at 3.80 GHz using 31 GB of RAM along with a GeForce GTX 1080 Ti GPU unit by using the `Tensorflow` package. The training phase firstly is carried on for 5000 epochs with the Adam optimizer [41] with learning rate $\lambda = 0.001$, and afterwards for 10,000 epochs with the L-BFGS-B algorithm [42], following the method originally proposed in [25] to have (empirical) convergence guarantees. The weights are initialized under the Xavier initialization method [43]. The adopted non-linear activation function for each layer is the hyperbolic tangent `tanh`. Given different initial $\mathbf{W}^0, \mathbf{b}^0$, it is observed that PINNs may converge to different solutions, see, e.g., [44,45]. Hence, our experiments are performed 10 times, changing the initial random seed and producing 10 approximate solutions. Since there is no guarantee of a unique solution, as a non-convex optimization problem is solved by minimizing (4), the selected $\widehat{\mathcal{M}}_{\boldsymbol{\theta}}$ which is retained corresponds to the solution achieving the smallest residual term.

The PINNs code outputs the predicted evaluation of $\widehat{\mathcal{M}}_{\theta}$ at points $(t_i, t_j)$ inside $\widehat{\Omega}$. At this stage, a quasi-interpolant spline*s* is constructed by adopting Formula (2).

The generation pipeline, outlined in Algorithm 1, can be summarized as follows:

- The boundary $\partial\Omega$ is split into 4 pieces $\Gamma_i$, for $i = 1, \dots, 4$, by performing for example knot-insertion.
- Each $\Gamma_i$ is then parametrized as a Bspline curve $g_i : [0, 1] \to \Gamma_i$.
- PINNs are trained to minimize the loss functional in Equation (4) over a set of boundary points and over the Laplace equation.
- The trained network $\widehat{\mathcal{M}}_{\theta}$ represents an approximation of the sought parameterization map $\mathcal{M}$.
- Uniformly spaced grid points are generated in $\widehat{\Omega}$ and mapped by $\widehat{\mathcal{M}}_{\theta}$ to $\Omega$.
- A continuous spline approximation of $\widehat{\mathcal{M}}_{\theta}$ is obtained by using a Hermite Quasi-Interpolation operator (QI).

---

**Algorithm 1** Pseudo-code for the proposed algorithm

---

**Data:** Given $\delta\Omega$
**Result:** Parameterization mapping $\mathcal{M}$ of $\Omega$
**begin**
    Split $\partial\Omega$ in 4 boundary curves $\Gamma_i$, $i = 1, \dots, 4$;
    Parametrize each $\Gamma_i$ with B-splines $g_i : [0, 1] \to \Gamma_i$, $i = 1 \dots, 4$;
    Set up PINNs architecture;
    $\widehat{\mathcal{M}}_{\theta} \longleftarrow$ Minimize $\mathcal{L}(\mathcal{T}, \theta)$ in Equation (4);
    Apply the chosen QI operator to $\widehat{\mathcal{M}}_{\theta}$ by using formula (2);
    *If needed, apply post-processing;*
**end**

---

## 4. Numerical Examples

In this section, some numerical experiments are performed on specific planar shapes which are typical benchmarks considered in assessing the quality of the produced parameterization, see, e.g., [7,15]. The experiments have been chosen with increasing complexity. Firstly a fully symmetric and convex domain is assayed. Then, for the second example, $\Omega$ is a non-convex and non-symmetric domain, but it can be obtained by a simple deformation of $\widehat{\Omega}$. The third benchmark is a slightly non-convex domain, while the fourth and fifth examples consider highly non convex and non symmetric shapes. Moreover, some comparisons are shown with respect to two techniques suitable for 4-sided shaped domains which only need a description of the boundary, Coons patches and the inpaint technique, which is computing a harmonic mapping $\mathcal{M}$ as introduced in (1), such that

$$\begin{cases} -\Delta\mathbf{x} = 0 & \text{in } \Omega \\ -\Delta\mathbf{y} = 0 & \text{in } \Omega \end{cases}$$

with boundary conditions given as in Problem 3. This mapping is computed by approximating the discrete Laplace operator by finite difference formulas. John D'Errico (2023). inpaint_nans (https://www.mathworks.com/matlabcentral/fileexchange/4551-inpaint_nans, accessed on 16 April 2023), MATLAB Central File Exchange. The evaluation of the three techniques is performed by checking if the fundamental bijectivity requirement ('Bij') is satisfied, namely, the determinant of the mapping should always be different from zero and never change sign. Then, the quality of the produced parameterization is also checked by computing the following functional,

$$W := \int_{\widehat{\Omega}} \frac{\|\frac{\partial\mathcal{M}}{\partial t_1}\|^2 + \|\frac{\partial\mathcal{M}}{\partial t_2}\|^2}{det J} \, dt_1 \, dt_2, \tag{5}$$

with $J$ denoting the Jacobian matrix of the mapping $\mathcal{M}$.

Expression (5) is known as the Winslow functional, see, e.g., [46] and in order to have a parameterization which is as conformal as possible, i.e., it should be almost a composition of a scaling and a rotation matrix, the best value for $W$ should be 2. Note that, in the specific settings, having a good value for $W$ is an additional property but it is not a necessity, as no additional constraints are imposed to guarantee the minimal achievable $W$.

### 4.1. Circle

As first example, a unit circle is considered. The boundary is sampled with 30 points on every $\Gamma_i$, $i = 1, \ldots, 4$ and the final residual term value is $\mathring{\mathcal{L}}(\mathring{\mathcal{T}}, \boldsymbol{\theta}) = 4.37 \times 10^{-5}$. The results with the three approaches are shown in Figure 2, where from top to bottom, on the left column, the distribution of the physical points inside $\Omega$ is shown and on the right column, the parametric lines of the computed parameterizations are plotted, for Coons, Inpaint, and Pinns methods, respectively.

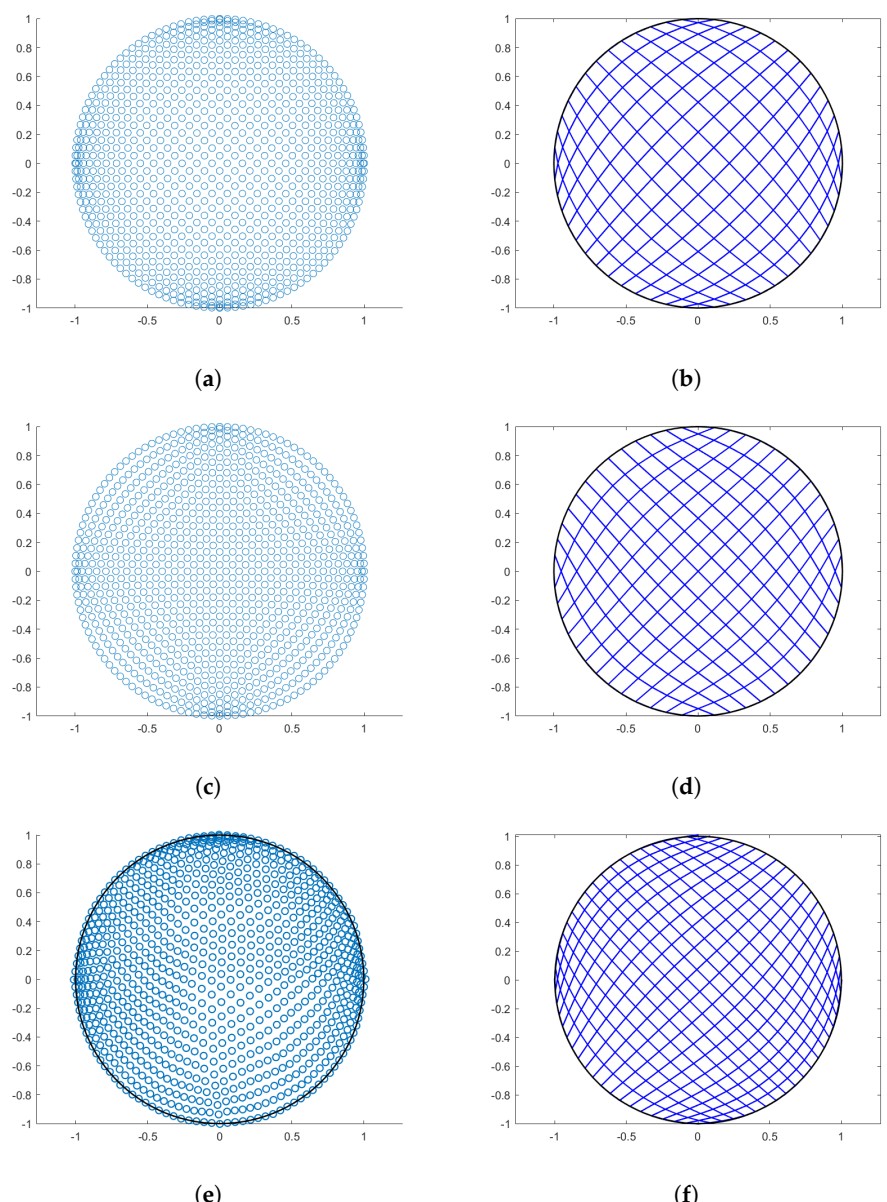

(a)

(b)

(c)

(d)

(e)

(f)

**Figure 2.** Circular domain. (**a**) Points generated via Coons; (**b**) Linear parameterization with Coons) (**c**) Points generated via inpaint; (**d**) Linear parameterization with inpaint; (**e**) Points with PINNs; and (**f**) PINNs-QI parameterization.

We remark that in this case the resulting parameterization is singular for all the methods at the image of the four corners of $\widehat{\Omega}$ and the considered shape is not particularly challenging, but this example is interesting for symmetry reasons and for visually appreciating how the three methods generate points inside the physical domain $\Omega$. Table 1 collects the evaluation results for this example. All the methods provide a bijective mapping, and, as expected, the values for $W$ are very close to the theoretical optimum.

**Table 1.** Circle-shaped domain evaluation.

| Method | Bij | W | min(det J) | max(det J) |
|:---:|:---:|:---:|:---:|:---:|
| Coons | yes | 2.1640 | 0.3150 | 4.7044 |
| Inpaint | yes | 2.1598 | 0.4141 | 3.7948 |
| PINNs | yes | 2.1639 | 0.3125 | 4.3160 |

*4.2. Wedge-Shape*

The second example shows a wedge-shaped domain, and the results are presented in Figure 3. In this case, 60 points per boundary curve are sampled and the final accepted mapping for the proposed approach has $\mathring{\mathcal{L}}(\mathring{\mathcal{T}}, \boldsymbol{\theta}) = 3.59 \times 10^{-6}$. All the three considered approaches perform well, as also indicated in Table 2.

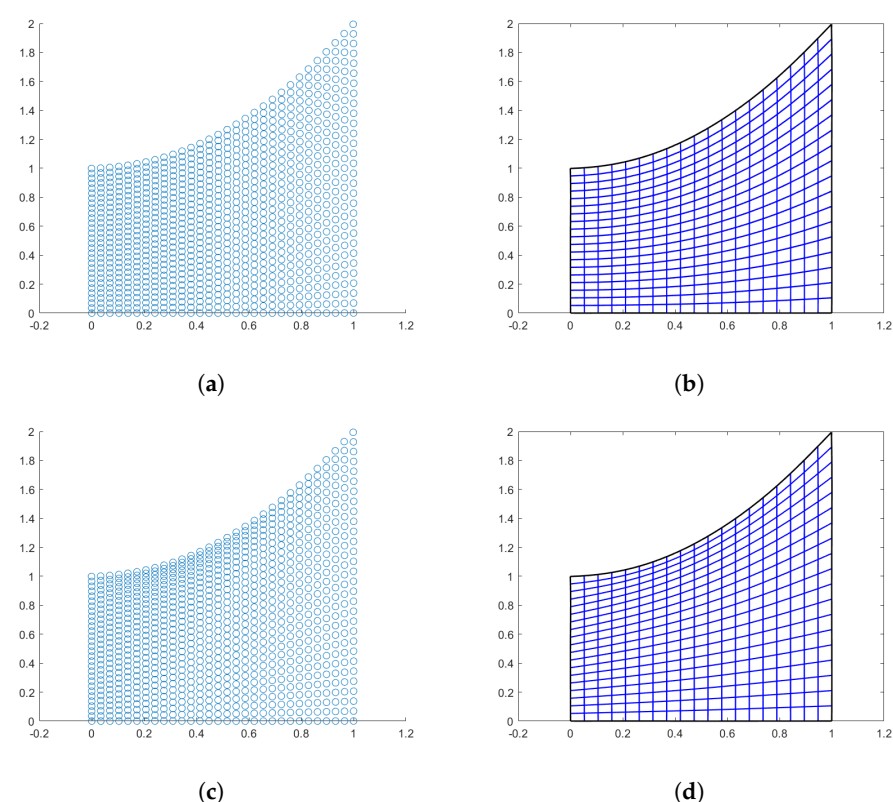

**(a)**　　　　　　　**(b)**

**(c)**　　　　　　　**(d)**

**Figure 3.** *Cont.*

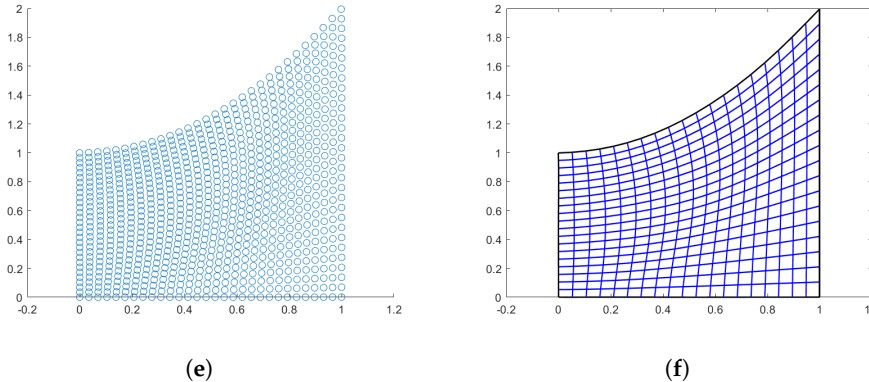

**Figure 3.** Wedge-shaped domain. (**a**) Points generated via Coons; (**b**) Linear parameterization with Coons; (**c**) Points generated via inpaint; (**d**) Linear parameterization with inpaint; (**e**) Points with PINNs; and (**f**) PINNs-QI parameterization.

**Table 2.** Wedge-shaped domain evaluation.

| Method | Bij | W | min(det J) | max(det J) |
|--------|-----|---|-----------|-----------|
| Coons | yes | 2.0834 | 0.9927 | 1.9635 |
| Inpaint | yes | 2.0812 | 0.6329 | 2.0284 |
| PINNs | yes | 2.0819 | 0.9928 | 1.8355 |

*4.3. Quarter-Annulus-Shaped Domain*

The next example is a quarter-annulus-shaped domain, with boundary curves $g_i$ expressed as B-splines with knot vectors (KV) $\Xi_1 = [0\ 0\ 0\ 1\ 1\ 1]$ and $\Xi_2 = [0\ 0\ 0\ 0\ 0.5\ 1\ 1\ 1\ 1]$ and control points $\mathbf{c}_i = (c_x, c_y)_i$ as given in Table 3, for $i = 1, \ldots, 4$.

**Table 3.** Descriptors for the quarter-annulus shaped boundary curves.

| Curve | KV | $c_x$ | $c_y$ |
|-------|-----|-------|-------|
| $g_1$ | $\Xi_1$ | $(-4, -2.5, -1)^T$ | $(0, 0, 0)^T$ |
| $g_2$ | $\Xi_2$ | $(-1, -1, -0.7, -0.4, 0)^T$ | $(0, 0.4, 0.7, 1, 1)^T$ |
| $g_3$ | $\Xi_1$ | $(0, 0, 0)^T$ | $(4, 2.5, 1)^T$ |
| $g_4$ | $\Xi_2$ | $(-4, -4, -4, -2, 0)^T$ | $(0, 2, 4, 4, 4)^T$ |

In Figure 4, we see the results obtained by the three methods with using linear splines interpolation for the Coons and Inpaint case, while using a quadratic quasi-interpolant spline for the PINNs method ($\mathring{\mathcal{L}}(\mathring{\mathcal{T}}, \theta) = 2.81 \times 10^{-5}$). Regarding the evaluation of this domain, the linear mapping Jacobian matrix results very ill conditioned, moreover, since $g_2$ and $g_4$ are cubic B-spline curves, it seems more pertinent to construct a cubic parameterization where the used knots for the boundary curves are a refinement of the vectors $\Xi_1$ and $\Xi_2$. Therefore, Table 4 shows the results obtained by adopting the same QI operator of bi-degree 3 on the given sample points provided by the three methods. The obtained mapping is bijective in all the three cases and the value for the Winslow functional is almost optimal as well.

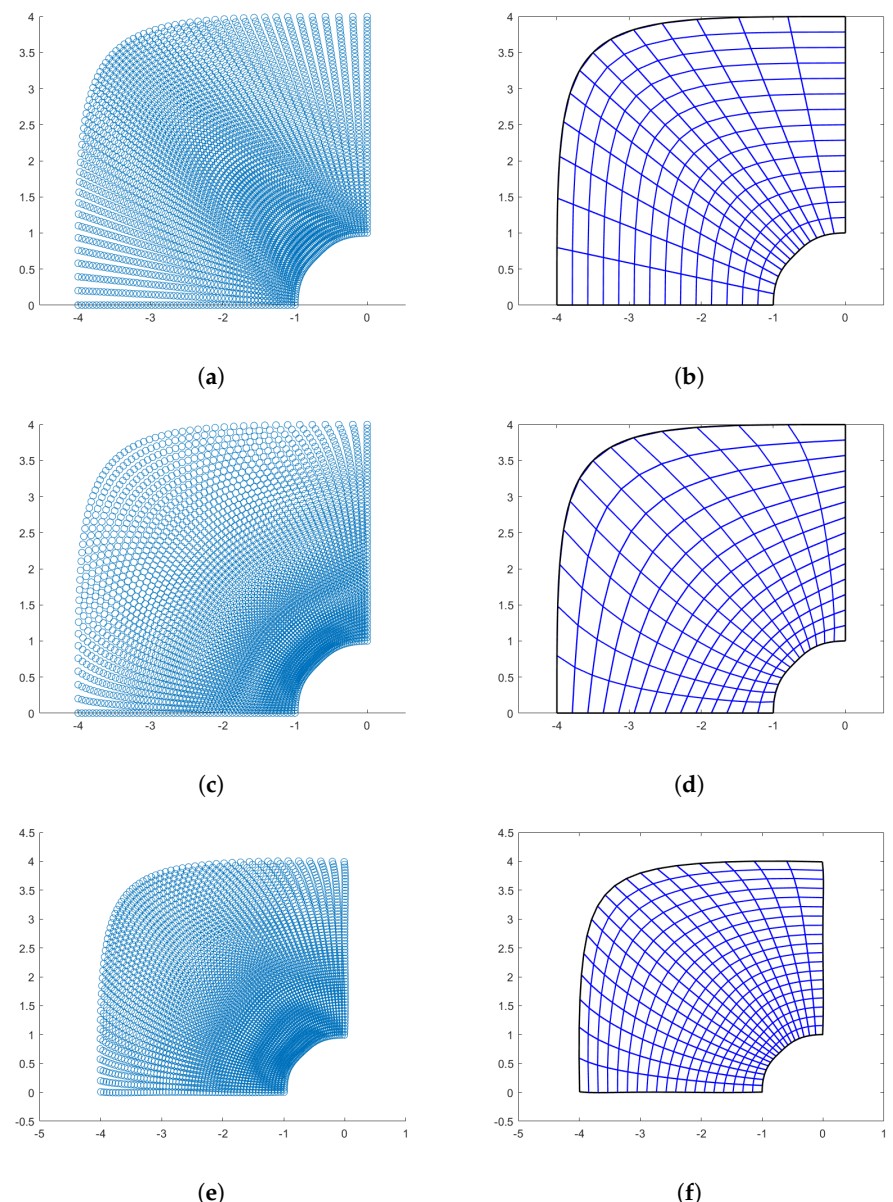

**Figure 4.** Quarter-annulus domain. (**a**) Points generated via Coons; (**b**) Linear parameterization with Coons; (**c**) Points generated via inpaint; (**d**) Linear parameterization with inpaint; (**e**) Points generated with PINNs; and (**f**) PINNs-QI parameterization.

**Table 4.** Quarter circle-shaped domain evaluation.

| Method | Bij | W | min(det J) | max(det J) |
|--------|-----|------|-----------|-----------|
| Coons | yes | 2.4242 | 5.0519 | 31.7624 |
| Inpaint | yes | 2.1631 | 2.1631 | 2.5388 |
| PINNs | yes | 2.2287 | 3.3262 | 31.1962 |

### 4.4. Hourglass-Shaped Domain

The next example is an hourglass-shaped domain with boundary curves given by cubic B-splines defined on the knot vector $\Xi = [0\,0\,0\,0\,0.5\,1\,1\,1\,1]^{\top}$ for $i = 1, \dots, 4$ and with corresponding control points reported in Table 5.

**Table 5.** Control points for the hourglass shaped domain.

| Curve | $c_x$ | $c_y$ |
|:---:|:---:|:---:|
| $g_1$ | $(1.5, 3.5, 5.6, 8, 10)^T$ | $(1.5, 2, 2.7, 2, 1.8)^T$ |
| $g_2$ | $(10, 7, 6, 7, 10)^T$ | $(1.8, 4, 7, 10, 13)^T$ |
| $g_3$ | $(1.2, 3.5, 5.6, 8, 10)^T$ | $(13, 12, 11.7, 12.5, 13)^T$ |
| $g_4$ | $(1.5, 4, 5, 4, 1.2)^T$ | $(1.5, 4, 7, 10, 13)^T$ |

In Figure 5, the hourglass-shaped domain is parametrized by using the three different techniques. On the first row the linear parameterization generated via Coons patches is shown. This parameterization although bijective, as highlighted in the zoomed frame, Figure 5b, and by the fact that its determinant is always strictly positive, presents highly distorted quadrilaterals in the center. On the second row, the parameterization produced by the Inpainting results to be singular, see Figure 5d. At the right boundary, the mapping fails to satisfy the prescribed boundary conditions. Finally, on the last row of Figure 5, the parameterization obtained with the proposed approach is shown. In this case, $\mathring{\mathcal{L}}(\mathring{\mathcal{T}}, \boldsymbol{\theta}) = 1.20 \times 10^{-4}$. In particular, since at the right boundary the produced quadrilaterals look slightly distorted, and the obtained mapping is no longer orientation-preserving, i.e., its determinant is changing sign, in order to improve the quality, a post-processing step, fully described in the next section, is added to the pipeline of the method.

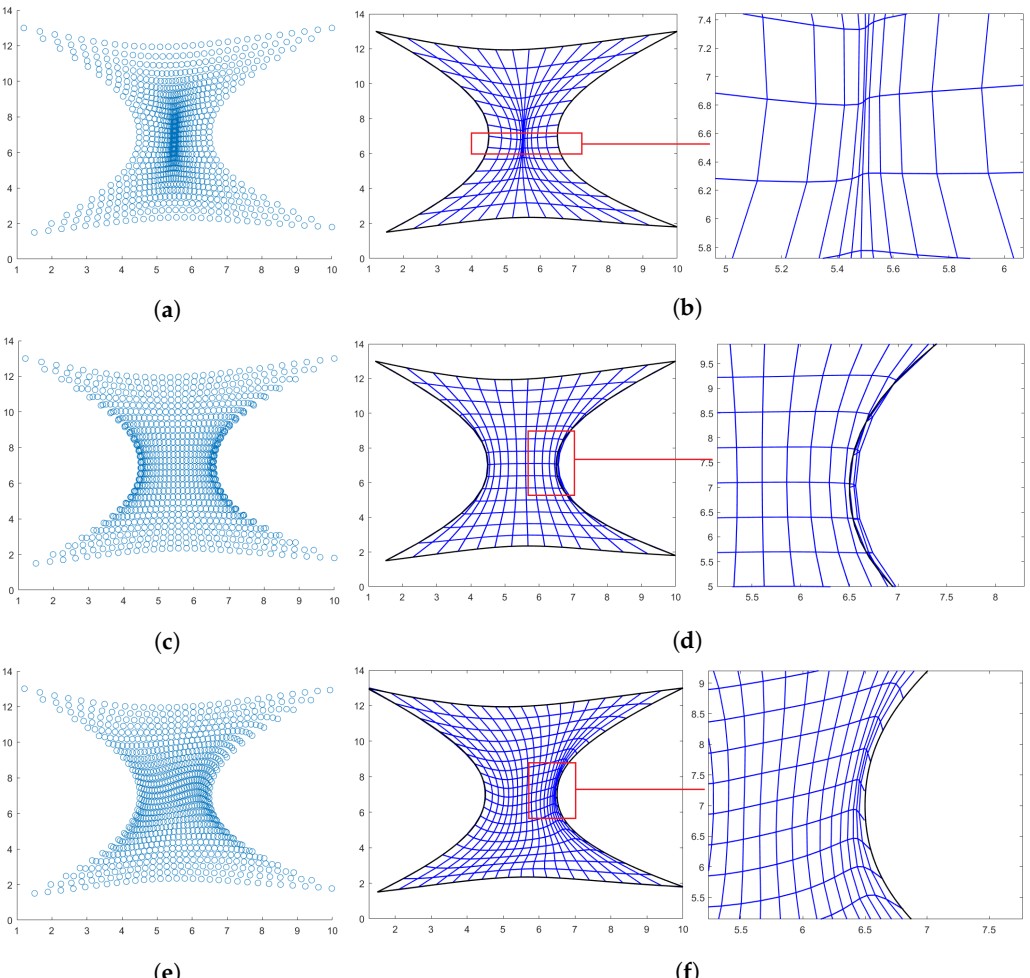

**Figure 5.** Hourglass-shaped domain. (**a**) Points generated via Coons; (**b**) Linear parameterization with Coons and zoom in; (**c**) Points generated via inpaint; (**d**) Linear parameterization with inpaint and zoom in (**e**) Points generated via PINNs; and (**f**) PINNs-QI parameterization and zoom in.

Furthermore, in this case, since the boundary curves are cubic B-splines, in order to exactly reproduce $\partial\Omega$, a bicubic QI spline is constructed on the given sample points for all the three methods. Table 6 reports the results. Moreover, in order to make clearer the behavior of the mapping, a surface plot of the determinant of the Jacobian matrix is also shown for all the three methods in Figure 6. Note that the high values for the $max(det J)$ should not cause too much concern, as they occur only at the corners of the domain, where the influence of such strong features reflects in high curvature of the produced mapping.

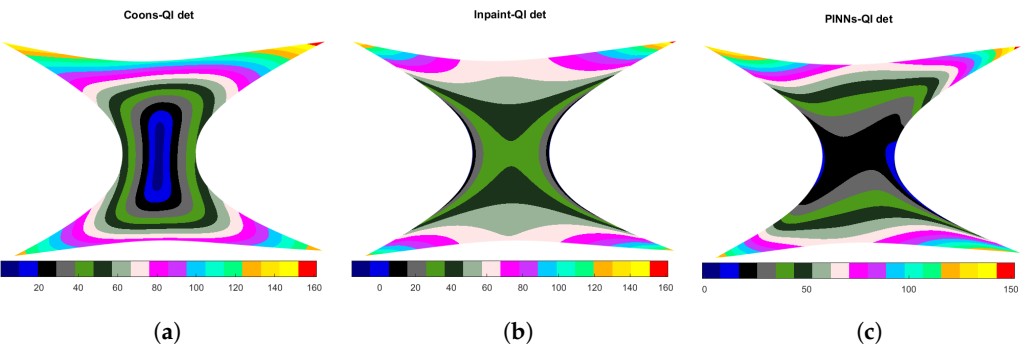

(a)     (b)     (c)

**Figure 6.** Determinant of the Jacobian matrix for the bicubic parameterization. (**a**) Method: Coons-QI; (**b**) Method: Inpaint-QI; and (**c**) Method: PINNs-QI.

**Table 6.** Hourglass-shaped domain evaluation.

| Method | Bij | W | min(det J) | max(det J) |
|---|---|---|---|---|
| Coons | yes | 6.7636 | 0.4713 | 161.3302 |
| Inpaint | no | 8.2491 | −15.7232 | 161.3302 |
| PINNs | no | 2.1853 | −1.1140 | 151.6974 |
| PINNs-Post | yes | 4.0696 | 4.7709 | 339.1136 |

### 4.5. Butterfly-Shaped Domain

The last example is a more challenging planar domain resembling a butterfly. This domain presents very sharp features which makes it a hard shape to parametrize especially near the four corners. The parameterization of the four boundary curves is realized with cubic B-spline functions with knot vectors, $\Xi_1 = [0\ 0\ 0\ 0\ 0.2\ 0.5\ 0.8\ 1\ 1\ 1\ 1]$ and $\Xi_2 = [0\ 0\ 0\ 0\ 0.5\ 1\ 1\ 1\ 1]$ and control points, as reported in Table 7.

**Table 7.** Control points for the butterfly-shaped domain.

| Curve | KV | $c_x$ | $c_y$ |
|---|---|---|---|
| $g_1$ | $\Xi_1$ | $(4, 5, 8, 9.2, 11, 14, 16)^T$ | $(1, 5, 5, 7, 5, 5, 1)^T$ |
| $g_2$ | $\Xi_2$ | $(16, 16, 13, 13, 17)^T$ | $(15, 7, 11, 15)^T$ |
| $g_3$ | $\Xi_1$ | $(1, 5, 7, 9, 11, 14, 17)^T$ | $(15, 13, 14.5, 12, 14, 13, 15)^T$ |
| $g_4$ | $\Xi_2$ | $(4, 3, 6, 6, 1)^T$ | $(1, 5, 7, 11, 15)^T$ |

The results obtained with the three methods are shown in Figure 7. None of the methods provides a bijective mapping due to some folding in the center for the Coons patches method and near the south boundary curve for the Inpaint technique. About PINNs ($\mathring{\mathcal{L}}(\mathring{\mathcal{T}}, \boldsymbol{\theta}) = 0.01627$) the main issues occur near the left-side boundary curve as the accuracy of the constructed net seems to become poorer near this side of the domain. Otherwise, everywhere else, inside $\Omega$, the produced mapping results perfectly injective. As in the hourglass-shaped domain, to have an exact boundary representation, a cubic QI spline is constructed for all the three methods. Due to the presence of self-intersections though, in some cases, the results for the Winslow functional are equal to $\infty$, see Table 8.

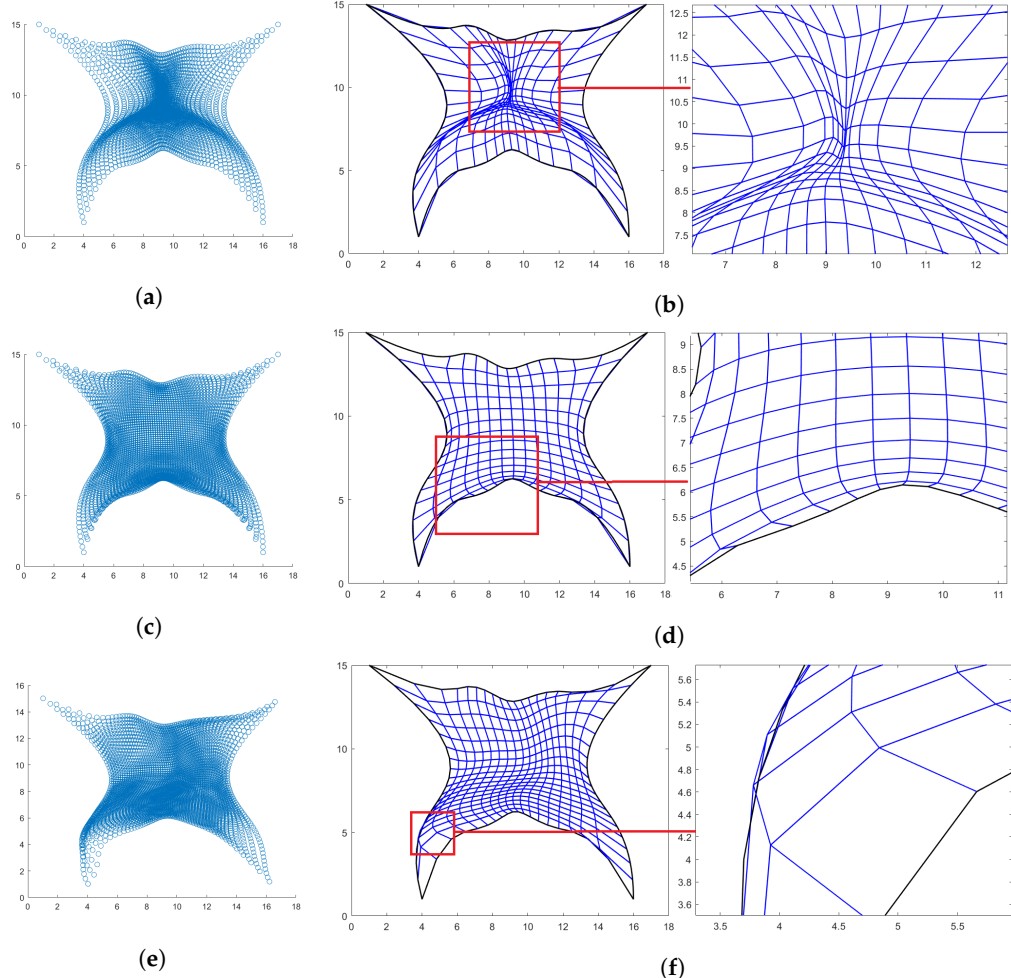

**Figure 7.** Butterfly-shaped domain. (**a**) Points generated via Coons; (**b**) Linear parameterization with Coons and zoom-in; (**c**) Points generated via inpaint; (**d**) Linear parameterization with inpaint and zoom-in; (**e**) Points generated with PINNs; and (**f**) PINNs-QI parameterization and zoom-in.

**Table 8.** Butterfly-shaped domain evaluation.

| Method | Bij | W | min(detJ) | max(det J) |
|--------|-----|---|-----------|------------|
| Coons | no | $\infty$ | $-51.7043$ | $1.0693 \times 10^3$ |
| Inpaint | no | $\infty$ | $-254.3629$ | $1.0695 \times 10^3$ |
| PINNs | no | 2.9064 | $-114.9197$ | $1.2973 \times 10^3$ |
| PINNs-Post | yes | 2.6513 | 0.045 | $4.3519 \times 10^3$ |

For the PINNs-QI method, although the mapping is not bijective as well, this happens in a very narrow stripe near the left boundary and, also thanks to the regularization provided by the QI, the Winslow functional result is not spoiled. Furthermore, to give a better idea where the mapping fails to be bijective, in Figure 8 the *detJ*, obtained with the three methods, is plotted as a surface. As for the previous example, the high values for the *max(detJ)* occur only at specific areas near the corners of the domain.

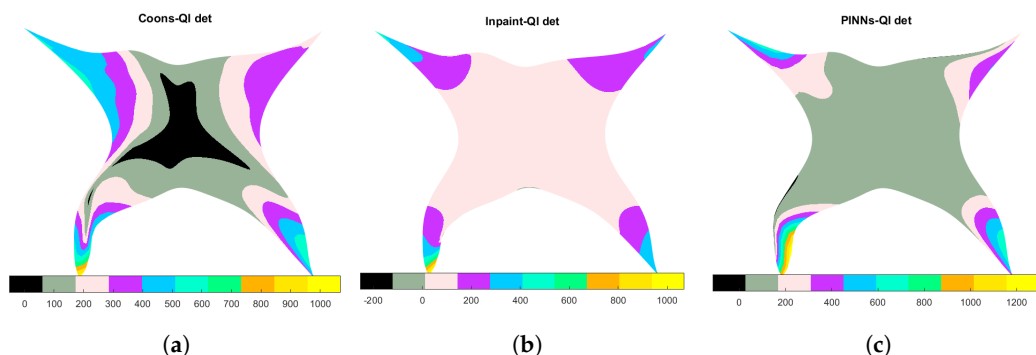

**Figure 8.** Determinant of the Jacobian matrix for the obtained parameterization. (**a**) Method: Coons; (**b**) Method: Inpaint; and (**c**) Method: PINNs-QI.

## 5. Post-Processing Correction

In order to improve the obtained parameterization when it is not orientation-preserving or when it is not bijective, we rely on the ability of the adopted QI to provide a correction in the following sense. In particular, the hourglass-shaped domain and the butterfly-shaped domain are the cases addressed here. For the first case, as is shown in the zoomed-in frame of Figure 5f and in Figure 6c, the Jacobian of the produced mapping is changing sign near the right-most side of $\Omega$. Hence, the second to last list of physical points outputted by the PINNs code is deleted and regarding the parameter domain, the corresponding parameters are also deleted, producing therefore a non-uniform grid near the right edge. The determinant of the Jacobian matrix of the new mapping ranges between 4.7709 and 339.1136, obviously the highest values occur at the right corners of the physical domain, as the produced quadrilateral cells are bigger. Furthermore, the value for the Winslow functional slightly increases to 4.0696, but this is not a significant price to pay to obtain an orientation-preserving mapping. These results are reported in the last row of Table 6 and the final parameterization with the determinant of its Jacobian matrix is shown in Figure 9.

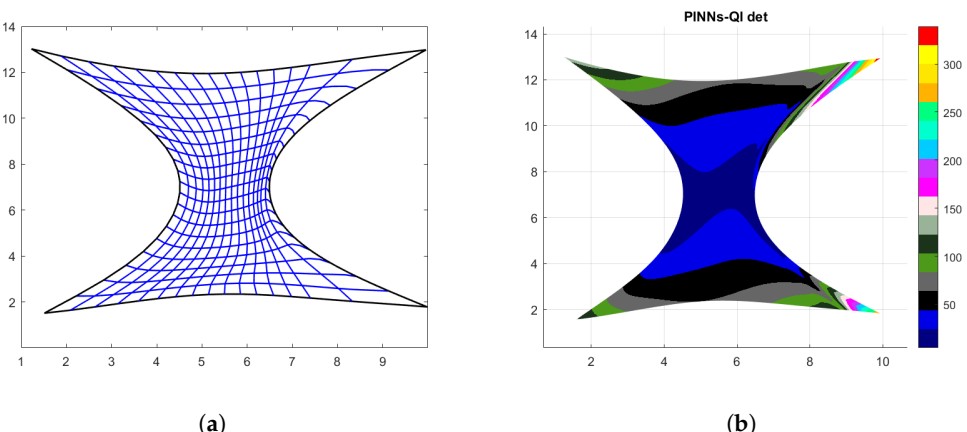

**Figure 9.** Correction output. (**a**) PINNS-QI parameterization after post-processing; and (**b**) Determinant of J for the post-procecessed mapping.

Regarding the butterfly-shaped domain, the main issues occur on the left edge of $\Omega$, see Figures 8c and 10a and also at top edge. Therefore, the faulty physical points are removed, as well as their corresponding parameters in $\widehat{\Omega}$, and the obtained results can be visually appreciated in Figures 10b and 11. The new values for *det J* varies between 0.045 and $4.3519 \times 10^3$ and the Winslow functional is 2.6513. These results are also reported in the last row of Table 8.

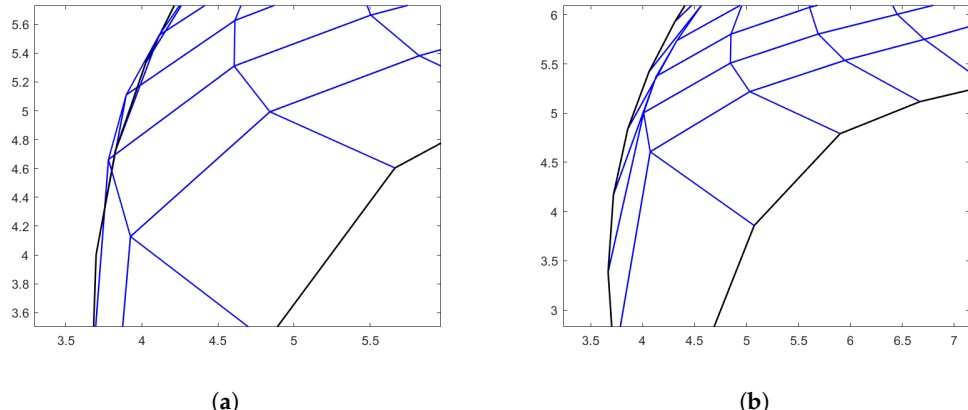

(**a**)　　　　　　　　　　　　　(**b**)

**Figure 10.** Comparison before and after post-processing. (**a**) Zoom-in where there is no injectivity; and (**b**) Zoom-in after the post-processing correction.

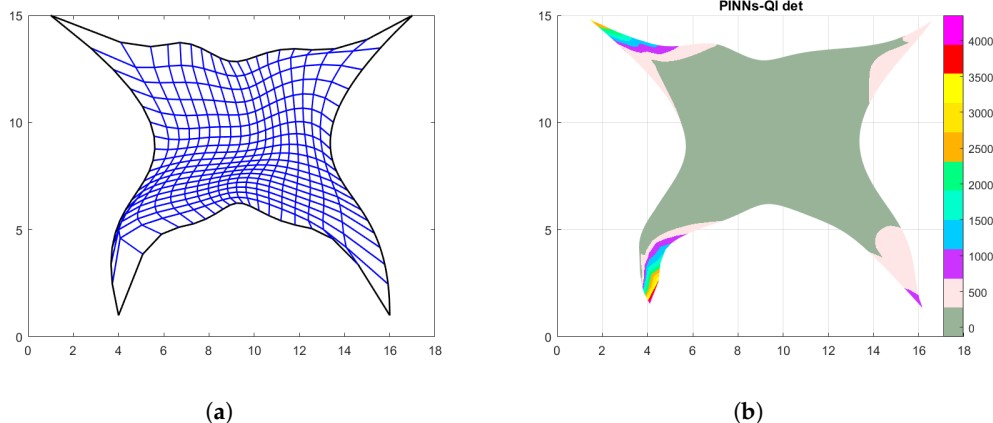

(**a**)　　　　　　　　　　　　　(**b**)

**Figure 11.** Results after the post-processing correction. (**a**) PINNs-QI parameterization after post-processing; and (**b**) Determinant of the Jacobian matrix after post-processing.

## 6. Conclusions

In the present paper we have shown how PINNs-driven methods can be profitably used to construct a parameterization of a planar domain $\Omega$ by only knowing its boundary representation. In more detail, once an internal set of physical points is obtained, quasi-interpolation is applied to generate a suitable spline parameterization of the given domain. Several planar shapes have been considered with increasing complexity and compared to existing techniques, such as Coons patches and EGG-based methods, needing only the description of the boundary of the computational domain; the obtained results show that this approach is promising. In particular, from the conducted experiments is evident how for symmetric and slightly non-convex domains, all the considered approaches perform well by producing a bijective parameterization which also achieves an almost optimal value for $W$ in all the cases. Regarding more complex shapes, i.e., non-symmetric and highly non-convex domains all the methods exhibit evident faults. Nevertheless, the PINNs-based approach is more robust than the one based on Coons patches, as the produced interior points never allow for folding or self-intersection of the mapping. Regarding the Inpaint technique, being also an EGG method, and, hence, driven by a similar approach, the results are more alike. The main difference in this case lies in the value for $W$; this is very similar for less complex shapes and much worse for the Inpaint technique when highly non convex domains are analyzed. Future work will be devoted to the study of more sophisticated loss functionals for the considered purpose by trying to achieve bijective mappings that have low distortions, see, e.g., [47,48] and to the extension to the non-planar case.

**Author Contributions:** Conceptualization, F.M. and M.L.S.; methodology, F.M. and M.L.S.; software, G.A.D. and A.F.; validation, G.A.D. and A.F.; writing—Review and editing, A.F., G.A.D., M.L.S. and F.M. All authors have read and agreed to the published version of the manuscript.

**Funding:** The research of Antonella Falini was funded by PON Ricerca e Innovazione 2014-202 FSE REACT-EU, Azione IV.4 "Dottorati e contratti di ricerca su tematiche dell'innovazione" CUP H95F21001230006. The research of Francesca Mazzia is funded under the National Recovery and Resilience Plan (NRRP), Mission 4 Component 2 Investment 1.4—Call for tender No. 3138 of 16 December 2021 of Italian Ministry of University and Research funded by the European Union—NextGenerationEU, Project code: CN00000013, Concession Decree No. 1031 of 17 February 2022 adopted by the Italian Ministry of University and Research, CUP: H93C22000450007, Project title: "National Centre for HPC, Big Data and Quantum Computing". The research of Maria Lucia Sampoli is co-funded by European Union—Next Generation EU, in the context of The National Recovery and Resilience Plan, Investment 1.5 Ecosystems of Innovation, Project Tuscany Health Ecosystem (THE), CUP: B83C22003920001.

**Data Availability Statement:** The developed codes can be released under specific request.

**Acknowledgments:** All the authors are member of the INdAM GNCS national group. Antonella Falini and Francesca Mazzia thank the GNCS for its valuable support under the INDAM-GNCS project CUP_E53C22001930001.

**Conflicts of Interest:** The authors declare no conflict of interest.

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
