# Peer review of "Splines Parameterization of Planar Domains by Physics-Informed Neural Networks"

_mathematics, doi:10.3390/math11102406_

Round 1
Reviewer 1 Report
1) All the acronyms must be written in full the first time they appear in-text
2)"In recent years, Physics Informed Neural Networks (PINNs) have been proved to be a powerful tool to compute the solution of PDEs replacing standard numerical models with
deep neural networks"
Mention some of these standard numerical models to make it clear to the readers
3) Towards the end of the abstract, the authors should provide some summary of the obtained results.
4) There is need to define all the variables in equation (2), for instance, x, y, t1 and t2 have not been defined.
5) Towards the end of Section 1, the authors should articulate the contributions of this paper (preferably in point form)
6)All the equations should be serialized for easy of reference.
7) The authors can incorporate a generalized pseudo-code that describes the basic operations of the proposed method.
8) "In this Section some numerical experiments are performed on specific planar shapes"
What informed the selection of Circle,Wedge-shape,Quarter-annulus shape domain,Hourglass shape domain, and Butterfly shaped domain?
9) "Several planar shapes have been considered and compared to existing techniques, such as Coons patches and EGG based methods: the obtained results show that this approach is promising"
This statement should be supported by briefly describing some of the obtained results here.
10) In which platform was the training and testing executed in?
11) "The training phase firstly is carried on for 5000 epochs with the Adam optimizer [39] with learning rate λ = 0.001, and afterwards for 10.000 epochs with the L-BFGS-B algorithm [40].."
i) How was this initial number of epochs and learning rate arrived at?
ii) What could be the effect of using lower or higher values for epochs and learning rate?
iii) Are the authors referring to 10,000 epochs or 10.000 epochs?
12) "The adopted nonlinear activation function for each layer is the hyperbolic tangent tanh". The rationale for the selection of this activation function need to be clarified.
13) Percentage improvements by PINNs can be included in all comparative tables
Moderate
Reviewer 2 Report
Please find my remarks below:
1. Please provide full form of abbreviated terms at the point of its first occurrence.
2. In general, the paper is difficult to read. In order to improve the readability of the paper, I suggest you to (a) provide a clear problem statement, (b) describe why this work is important, (c) formulate research questions, and (d) design experiments to investigate the research questions.
3. There is a significant gap on page 5. Please take necessary corrective action.
4. Compare your work with other relevant work that exists in the literature.
In general, the quality is good. Minor editing of English language required
Reviewer 3 Report
This paper addresses the fundamental problem (planar domain parameterization) of isogeometric analysis, this is done by using the Physics Informed Neural Networks model together with the QI-Hermite technique. Several tests are presented to validate the effectiveness of the proposed method. The research topic is important and the technical contribution is adequate. I prefer to recommend the acceptance provided the following comments are dealt with appropriately.
1. I am concerned about the bijectivity of the proposed approach. Is there any theoretical guarantee about this?
2. What is the purpose of presenting the circle-shaped domain and the quarter circle-shaped domain? The data shown in Table 1 and Table 4 indicates that the method “Inpaint” produces better results in these two examples than the new method.
3. The reference list is almost adequate, but I would suggest to consider also other papers focusing on constructing high-quality (bijective and low-distortion) parameterizations.
Reviewer 4 Report
- The presentation of the numerical method based on basic spline function is not clear. It is better to explain the motivation of the suggested in steps.
- It is important to discuss the convergence and stability of the presented numerical method.
- The results were compared with other existing methods depending on the value of winslow function w=2, which is given in equation 6. They are not discussed in the given Tables . From the Tables, we see that the method Inpaint is the best depending on the value w=2.
- In the last two columns of the Tables, It is listed the two values min(det J) and max(det J). what effect dose it have on the results.
-
-
Minor editing of English language required.
Reviewer 5 Report
The work applies the newly developed approach of the so-called Physics Informed Neural Networks (PINNs) to solving the partial differential equation of elliptic type with the Dirichlet boundary conditions. The work is interesting, innovative, well-written, mathematically sound, and can be useful for various applications.
Several remarks on the paper.
In the numerical examples of several planar shapes, three approaches of Coons, Inpaint, and PINNs parameterizations are used, and the results are shown in tables with the values of Winslow (W), and min and max of det J. However, W values are mostly very close between the methods and to the optimal value, and the min and max values are given without discussion. The paper needs more consideration on which method is better or worse in each case, and how to find the best parameterization. More could be elaborated on the criteria for estimation and identification of the best solution.
Also, all acronyms should be defined from the first appearing, so the readers would not guess what exactly those could mean, even in the cases of the names such as PDE or CAD.
The link given in the footnote in p. 5 does not lead to the source.
Resuming, subject to the minor revision, the paper can be recommended for publication.
some suggestions are given in the comments to the authors.
Round 2
Reviewer 2 Report
Thank you for the revised version. I am happy with the changes made.
Reviewer 4 Report
The answer to comment 3 is unsatisfactory.